# Morphology and Molecular Phylogeny of Genus *Oedogonium* (Oedogoniales, Chlorophyta) from China

**DOI:** 10.3390/plants11182422

**Published:** 2022-09-16

**Authors:** Qian Xiong, Yangliang Chen, Qingyu Dai, Benwen Liu, Guoxiang Liu

**Affiliations:** 1Institute of Hydrobiology, Chinese Academy of Sciences, Wuhan 430072, China; 2College of Advanced Agricultural Sciences, University of Chinese Academy of Sciences, Beijing 100039, China

**Keywords:** Oedogoniales, morphological characteristics, molecular phylogeny, *Oedogonium*, paraphyletic group, basal cell shapes

## Abstract

Oedogoniales comprises the three genera *Oedogonium*, *Oedocladium*, and *Bulbochaete*, which include more than 600 described species. The classification of Oedogoniaceae is currently based on morphology, and the complicated morphological characteristics make species identification difficult, with the limited molecular data also restricting the phylogenetic analysis. In the present study, we collected 47 *Oedogonium* specimens from China and sequenced 18S rDNA, ITS2, ITS (ITS1 + 5.8S + ITS2), and rbcL sequences to conduct phylogenetic analyses. We selected nine morphological characteristics, most of which were considered important in traditional systematics, for comparison with the molecular phylogeny results. All the topologies based on different datasets showed similar results; *Oedogonium* was a paraphyletic group, and *Oedocladium* and *Bulbochaete* clustered with *Oedogonium*. The morphological characteristics matching the phylogenetic results showed that the types of sexual differentiation, characteristics of the oogonium (including shape, types of aperture, and ornamentation of oospore wall), division types of antheridial, and number of sperm of each antheridial, which are considered the most important morphological characteristics in traditional taxonomy of *Oedogonium*, did not form monophyletic lineages respectively, indicating that traditional systematics may not reflect the real phylogeny of the genus *Oedogonium*. In addition, a new taxonomical classification of the genus *Oedogonium* was presented according to the shapes of basal cells, which matched well with the phylogenetic topologies. In addition, we propose to divide the genus *Oedogonium* into two sections, section *Globosum* and section *Elongatum*, representing the species with spherical or sub-hemispherical basal cells and elongated basal cells, respectively.

## 1. Introduction

The order Oedogoniales belonging to the family Oedogoniaceae Chlorophyceae, Chlorophyta, includes three genera, *Oedogonium* Link ex Hirn, *Oedocladium* Stahl, and *Bulbochaete* Agardh [1,2,3,4]. More than 600 species have been described in this order, most of which can be found in fresh water throughout the world. Genus *Oedogonium* includes 444 species and 349 lower taxonomical units, genus *Bulbochaete* includes 113 species and lower taxonomical units, and genus *Oedocladium* includes 15 species [5]. The presence of branches and hairs are useful characteristics for distinguishing taxa at the genus level. *Oedogonium* has simple unbranched filaments; *Oedocladium* has branched filaments; and *Bulbochaete* forms bulb-based hairs [4,5,6,7,8,9,10,11,12,13]. Asexual reproduction occurs via zoospores with a complex flagellar apparatus that is formed by vegetative cells and germinate almost immediately. Sexual reproduction is by oogonia and spermatozoids; oogonia are single or in groups, arising as a result of division of a vegetative cell, opening by a pore or a split, through which the spermatozoid may pass; the oogonium when fertilized becomes the oospore, with a wall of one to three layers, which after a period of rest produces four zoospores [2,4,14,15,16].

During the past few years, phycologists have used different characteristics as criteria for dividing the genus *Oedogonium*. Sexual differentiation could be the criterion for taxonomical classification for all species of *Oedogonium*, according to the position and relation of the oogonium and the antheridia could be distinguished as monoecious; dioecious, macrandrous; dioecious, and nannandrous [6,10]. Gauthier recognized that the different types of oogonium apertures (pore or circumcision) should be the first taxonomic characteristic for the genus *Oedogonium*. Mrozińska [17,18] proposed that the number of spermatozoids of the antheridial cell should be chosen as the criterion in taxonomical classification below the rank of genus for *Oedogonium*.

With the development of molecular phylogenetics, phycologists later conducted a series of phylogenetic studies of Oedogoniales. Oedogoniales are monophyletic, and *Bulbochaete* may be a sister to the other two genera [19,20,21,22]. Using nuclear 18S rDNA of 10 *Oedogonium* species, Alberghina et al. [23] suggested that *Oedogonium* might not be monophyletic and that the morphological characteristics may not define the phylogenetic groups. Using 18S rDNA sequences, Mei et al. [24] found that *Oedocladium* formed a separate clade within *Oedogonium*, whereas *Bulbochaete* was relatively distant from the other two genera. Xiong et al. [25,26,27,28] indicated that *Oedogonium* was paraphyletic based on chloroplast genome protein-coding genes [25,26], mitogenome protein-coding genes [27], and the single-copy orthogroups of the transcriptomes [28].

Even though the genus *Oedogonium* has described a large amount of species based on morphology, the molecular phylogenetic studies included relatively limited samples, the molecular data in the public database about the order Oedogoniales are also limited, and the phylogeny of *Oedogonium* remains problematic. In addition, the relationships that the morphological characteristics used as criterion of *Oedogonium* with the molecular phylogenetic result have not been discussed comprehensively; more molecular data and a broader sampling is required for further study of this group.

In the present study, a total number of 47 *Oedogonium* specimens from China were sampled and the 18S rDNA, ITS, and nucleotide sequences of the chloroplast genome (cpDNA) rbcL gene of the majority of them were obtained for phylogenetic analyses. The relationships between the different morphological characteristics and phylogenetic results were comprehensively discussed. The objective of the present study was to explore the phylogenetic relationship of the genus *Oedogonium* based on additional taxa and to search for the morphological characteristics which could reflect the phylogeny of *Oedogonium*.

## 2. Results

### 2.1. Morphological Observation of Voucher Specimens

Characteristics of the *Oedogonium* taxa included in this study are listed in Table 1. Morphological observations of these taxa were based on materials from the field because taxonomic identification of Oedogoniales species requires analysis of the reproductive structures for sexual reproduction. Inability to define taxa as concrete species was primarily due to a lack reproductive structures or limited samples. We were unable to induce sexual reproduction, but limited characteristics, such as shapes of vegetative cells, basal cells, and terminal cells, are listed.

### 2.2. Molecular Phylogeny

In the present study, we generated phylogenetic analyses by BI and ML methods based on ITS (ITS1 + 5.8S + ITS2), ITS2, and 18S + ITS (ITS1 + 5.8S + ITS2) + rbcL of 47 *Oedogonium* taxa newly sampled together with the sequences downloaded from the NCBI database. The ITS dataset consisted of 375 bp, of which 42.9% was variable informative and 32.5% was parsimony informative. The ITS2 sequence consisted of 213 bp, of which 67.61% was variable informative and 54.00% was parsimony informative. The concatenated dataset consisted of 3159 bp, of which 17.6% was variable informative and 9.5% was parsimony informative. The phylogenetic trees based on ITS sequences by both ML and BI showed the same results: separation of Oedogoniales into two clades with high support value (1/100 in BI and ML, respectively) (Figure 1). In the first clade, *Bulbochaete* clustered with *Oedogonium* and *Oedocladium* was separated into two clades, with Oe. prescottii clustering with *Oedogonium* and the four Oe. carolinianum strains forming the other clade; the remaining species of *Oedogonium* formed the second clade. We also used the ITS2 sequence with 213 bp to reconstruct the phylogenetic tree (Appendix A), and the general topology was almost consistent with the result based on the concatenated dataset by ITS1, ITS2, and 5.8 S. We tried to concatenate the 18 S rDNA, ITS, and rbcL of 44 Oedogoniales taxa including 40 taxa of *Oedogonium* and 4 taxa of *Oedocladium* to reconstruct the phylogenetic relationship. The topologies were basically identical by BI and ML analyses that *Oedogonium* formed three main clades (Figure 2). In the second clade, species of *Oedocladium* clustered with the small clade formed by *Oedogonium* species. In addition, the other two clades were both composed of *Oedogonium* taxa.

Since all the topologies based on different datasets showed similar results, the phylogenetic tree based on ITS sequences including more species with complete morphological characteristics were used for the following analysis. The nine morphological characteristics (types of sexual differentiation; types of oogonium aperture; ornamentation of the oospore wall; division types of the antheridial cell; number of sperm of each antheridial cell; shape of the vegetative cell; shape of the basal cell; and shape of the terminal cell) were selected to match the phylogenetic tree based on ITS sequences to search for the characteristic reflecting the phylogeny of *Oedogonium* (Figure 3 and Appendix A). Our results showed that monoecious; dioecious, macrandrous; and dioecious, nannandrous *Oedogonium* taxa did not form monophyletic lineages, and, similarly, the type of oogonium aperture; type of oogonium opening; division type of the antheridial cell; number of sperm of each antheridial cell; shape of the vegetative cell; and shape of the terminal cell did not form monophyletic lineages. Instead, Oedogoniales was separated into two clades according to the shape of the basal cell (elongated or spherical and sub-hemispherical) with a strong supported value. In the first clade, the taxa possessed elongated basal cells, and in the second clade, they possessed spherical or sub-hemispherical basal cells (Figure 3). Hence, we suggested that the genus *Oedogonium* can be divided into two sections based on the two kinds of basal cell shapes, namely section *Globosum* and section *Elongatum*, representing the species with spherical or sub-hemispherical basal cells and elongated basal cells, respectively. The descriptions of the two sections as follows: 

*Globosum* sect.

The basal cell spherical or sub-hemispherical; the filaments are relatively small with the width almost less than 10 um; the oogonium operculate, most division median, smooth oospore wall. 

Type species: *Oedogonium* capitellatum Wittrock ex Hirn (1900, 149, pl. XXIII). 

*Elongatum* sect.

The basal cell is elongated, and the other characteristics are the same as the description of genus *Oedogonium* (Link ex Hirn, 1900). 

Type species: *Oedogonium* grande Kützing ex Hirn (1900, 143, pl. XXI).

## 3. Discussion

In the present study, the phylogenetic results based on ITS, ITS2 and 18 S + ITS + rbcL by both BI and ML methods showed that *Oedogonium* was polyphyletic, and *Oedocladium* was separated into two clades clustering with *Oedogonium*, which were identified with our previous studies [25,26,27,28], and both *Bulbochaete* and *Oedocladium* clustered with *Oedogonium*. *Oedocladium* is not considered an independent lineage; this was explained by studies by Liu et al. [13], in which *Oedogonium* pakistanense, one of the few terrestrial species belonging to the genus *Oedogonium*, showed apical growth considered as a typical characteristic of *Oedocladium*. The authors proposed that *Oedogonium* pakistanense represents an evolutionary transition between *Oedogonium* and *Oedocladium* [13]. Based on our morphological observations with indoor cultures and field samples of *Oedogonium*, apical growth also occurs in many aquatic species, which showed a cladogenetic pattern and obscured phylogenetic signals between the two genera. Even though exact topologies differed based on ITS, ITS2 and 18 S + ITS + rbcL, they showed a basal clade including identical taxa (clade II and clade III, respectively).

Phylogenetic analysis combined with morphological studies revealed the morphological characters (the types of sexual differentiation; the characters of oogonium, including the shape, the types of the aperture, and the ornamentation of oospore wall; the division types of antheridial and the number of sperm of each antheridial) important in the traditional taxonomy of *Oedogonium* did not form monophyletic lineages, respectively, implying that the traditional morphological character used for the taxonomy of Oedogoniales may not reflect the real phylogeny of this group. The phenomenon of different types of sexual differentiation evolving multiple times has also been reported in bryophytes and plants [29,30,31,32,33,34,35,36,37]. In addition to these morphological characteristics recognized as important in traditional systematics, we also discussed vegetative, terminal, and basal cell shapes. Vegetative and terminal cell shape were not consistent with an evolutionary relationship for poor matching. Most of the species of Oedogoniales possessed cylindrical vegetative cells; a minority presented undulate or nodulose, punctate or granulate, distinctly capitellate and subhexagonal or subellipsoid shapes. In the present study, we collected the taxa with cylindrical and distinctly capitellate shapes to further evaluate this characteristic; more species with other types are needed for further analysis. Terminal cell shapes showed no accordance with evolutionary relationships mainly for the different environmental conditions; the same species may show different types of terminal cells according to the studies of Jao [10] and our observation during the past few years; for example, *Oedogonium* pringsheimii presented with obtuse or apiculate terminal cells in the field or in culture. However, basal cell shapes were found to match well with both of the topologies based on ITS, ITS2, and 18 S + ITS + rbcL, suggesting that basal cell shape is more consistent with the molecular phylogeny results. The basal cell of the filament of *Oedogonium* is most frequently elongated, and the attached lower end may be simple or lobed. In addition, the developing basal cell of the other *Oedogonium* species may be flattened into a subhemispherical cell or rarely appear nearly spherical. Based on samples of Jao [10] collected from a large area in China, *Oedogonium* included about 440 species, and the number of the species with subhemispherical or nearly spherical basal cells was 33. According to the statistics of the morphological characteristics of these 33 species, we found that they included the three kinds of sexual differentiation types, which showed again that the sexual differentiation types could not reflect the real phylogeny of the genus *Oedogonium*. In addition, all the 33 species also showed some common characteristics; for example, the filaments are relatively small, with the width almost less than 10 um; and with the oogonium subdepressed-globose; circumcision; and smooth ospore wall, and these characteristics were considered more primitive in the studies based on phenetic- and cladistics taxonomy methods by Mrozińska [17,18], implying that these features may be genetic linkage.

## 4. Materials and Methods

### 4.1. Sampling, Morphological Observation, and Culture Procedures

The 47 specimen strains described in this study were isolated from water samples and deposited in the Freshwater Algae Culture Collection at the Institute of Hydrobiology (FACHB collection), Wuhan, Hubei province, China. Voucher numbers are shown in Appendix A. Samples were isolated from the field by the authors and primarily examined under a stereoscope microscope and an Olympus BX53 (Tokyo, Japan) light microscope equipped with an Olympus DP80 digital camera. CellSens standard image analysis software (Tokyo, Japan) was used for morphological examination. Characteristics of the 47 specimens are summarized in Table 1.

### 4.2. DNA Extraction, PCR Amplification, and Sequencing

To obtain unialgal cultures, clean filaments were selected and transferred onto solid BG11 medium (1.5% BG11 agar) once or twice. All strains were grown at 20–25 °C in solid BG11 medium under a 12 h-12 h light-dark cycle at an intensity of 15–30 μmol/(m^2^·s). Once a unialgal culture was obtained, total genomic DNA was extracted using a Universal DNA Isolation Kit (AxyPrep, Suzhou, China) according to the manufacturer’s instructions. Polymerase chain reaction (PCR) amplification of partial 18 S rDNA, whole ITS rDNA regions, and rbcL were performed using 5 µL template DNA, 0.1 µM of each primer, and 25 µL 2× Tap Master Mix (ExTaq; Takara) in a 50 µL reaction volume. Nuclear-encoded 18 S rDNA sequences were amplified using the primers 18 F (5′-TGGTTGATCCTGCCAGT-3′) and 18 R (5′-TGATCCTTCTGCAGGTTCACC-3′; [38]). The amplification conditions were as follows: 5 min at 94 °C, 35 cycles of 50 s at 94 °C, 60 s at 55 °C, and 90 s at 72°C, and a final 10 min extension step at 72 °C. The ITS sequence was amplified using the primer pair designed by Primer5 ITSA1 (5′-ATGCTTAAGTTCAGCGGGTAG-3′) and ITSS1 (5′-GAACCTGCGGAAGGATCA-3′). The amplification conditions for ITS were as follows: 5 min at 94 °C, 32 cycles of 50 s at 94 °C, 60 s at 52 °C, and 90 s at 72 °C, and a final 10 min extension step at 72 °C. The rbcL sequence was amplified using the primer pair designed by Primer5 rbcL A3 (5′-CGATTAACTTATTATACGCC-20) and rbcL S3 (50-AGTTCTGGAGACCATTTG-18). The amplification conditions for rbcL were as follows: 5 min at 94 °C, 35 cycles of 50 s at 94 °C, 50 s at 56.5 °C, and 70 s at 72 °C, and a final 10 min extension step at 72 °C. PCR products were sequenced by TSINGKE Biotechnologies (China) and assembled using Seqman [39] and deposited in GenBank under the accession numbers provided in Appendix A.

### 4.3. Molecular Phylogenetic Analyses

The nuclear-encoded 18 S rDNA and ITS (ITS1 + 5.8 S + ITS2) sequences, and nucleotide sequence of the cpDNA rbcL gene were used for phylogenetic analyses. Sequences of related species were obtained from GenBank (Appendix A). In addition, 18 S rDNA, ITS, and ITS2 sequences were aligned using MAFFT 7.0 [40] and adjusted manually using MEGA7 [41]. The alignment result of the ITS sequences was shown in Appendix A. For nucleotide sequences of rbcL, the genes were additionally aligned using the MUSCLE function of MEGA7 with the option “align codons” [41,42] and were adjusted manually using MEGA7 [41]. Phylosuite [43] was used to concatenate 18 S rDNA, ITS, and rbcL sequences. Phylogenetic calculations were performed using maximum likelihood analysis (ML) by RAxML v8.2.10 [44]. A bootstrap analysis with 1000 replicates of the dataset was performed with ML to estimate statistical reliability. MrBayes v3.2.6 [45] was used for Bayesian inference (BI) with modeling of the GTR + I + G suggested by jModelTest2 [46]. Markov chain Monte Carlo (MCMC) analyses were run with four Markov chains (three heated, one cold) for 3,000,000 generations, and trees were sampled every 1000 generations. In each round of calculation, a fixed number of samples (burn-in = 1000) was discarded from the beginning of the chain.

## Figures and Tables

**Figure 1 plants-11-02422-f001:**
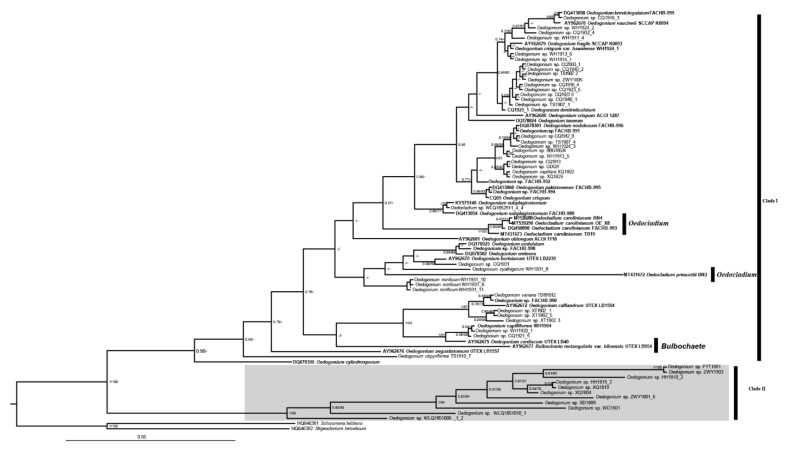
Phylogenetic tree of the Oedogoniales algae based on 80 ITS (ITS1 + 5.8 S + ITS2) sequences. Numbers on the left and right side of the branches represent ultrafast bootstrap inferred by Bayesian posterior probabilities (≥0.5) and RAxML (≥50%), respectively. Branch lengths are proportional to the genetic distances, which are indicated by the scale bar. The sequences downloaded from NCBI are marked in bold.

**Figure 2 plants-11-02422-f002:**
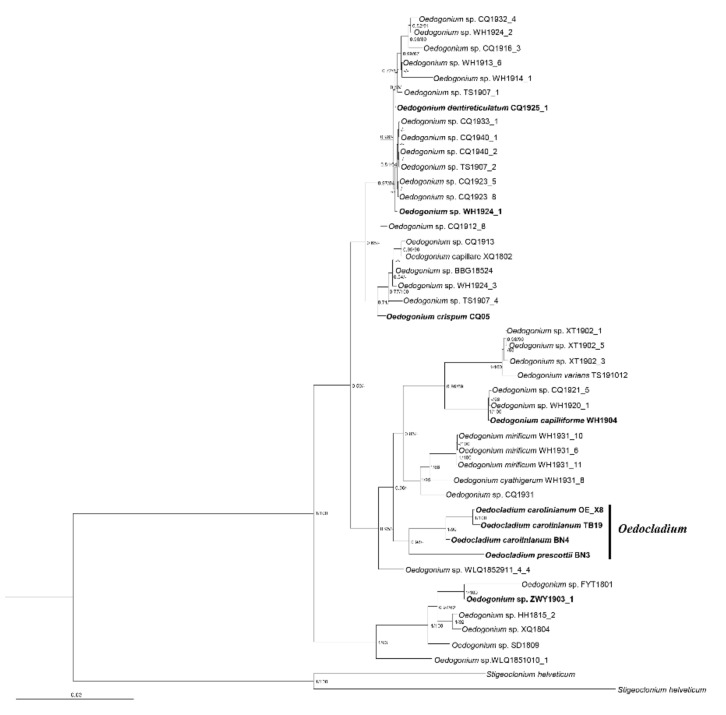
Phylogenetic tree inferred from 44 18 S, ITS1 (ITS1 + 5.8 S + ITS2) rDNA and *rbc*L sequences. Numbers on the left and right side of the branches represent ultrafast bootstrap inferred by Bayesian posterior probabilities (≥0.5) and RAxML (≥50%), respectively. Branch lengths are proportional to the genetic distances, which are indicated by the scale bar. The sequences downloaded from NCBI are marked in bold.

**Figure 3 plants-11-02422-f003:**
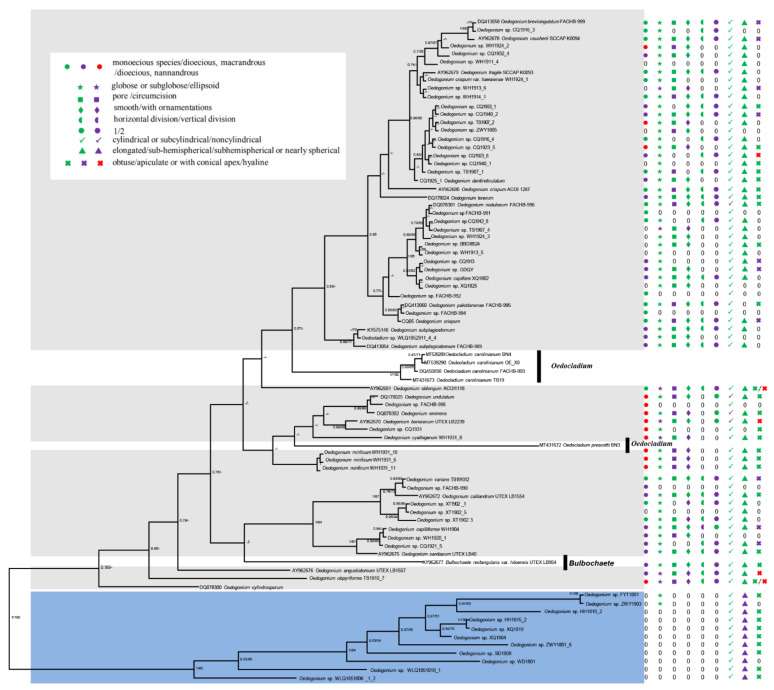
Phylogenetic tree inferred from 80 ITS (ITS1 + 5.8 S + ITS2) rDNA sequences with morphological traits. Numbers on the left and right side of the branches represent ultrafast bootstrap inferred by Bayesian posterior probabilities (≥0.5) and RAxML (≥50%), respectively. The main morphological traits including sex differentiation, the shape of oogonium, oogonium aperture, oospore wall ornamentation, the division way of antheridium, the number of sperm of each antheridium, vegetative cell, basal cell, and terminal cell displayed and represented by circles, asterisks, rectangles, lozenges, semi-circles, hexagons, triangles, ticks, and crosses, respectively.

**Table 1 plants-11-02422-t001:** Matrix of phenotypic traits scored for the five Oedogoniales strains. Character state definitions are below; unknown character states are notated as ‘‘0′’. Polymorphic conditions are indicated with multiple state numbers.

Terminal Cell: Obtuse (1); Apiculate or with Conical Apex (2); Hyaline (3); Taxa	Phenotypic Traits								
	1	2	3	4	5	6	7	8	9
***Oedogonium* sp. BBG18524**	0	1	1	1	0	0	1	1	1
***Oedogonium crispum* CQ05**	1	1	2	1	1	2	1	1	2
***Oedogonium* sp. CQ1912_8**	1	1	0	0	1	2	1	1	0
***Oedogonium* sp. CQ1913**	2	1	0	0	0	0	1	1	2
***Oedogonium* sp. CQ1916_3**	1	1	0	0	1	2	1	0	0
***Oedogonium* sp. CQ1916_4**	1	1	0	0	1	2	1	1	0
***Oedogonium* sp. CQ1921_5**	2	1	0	0	1	2	1	1	2
***Oedogonium* sp. CQ1923_5**	3	1	1	2	0	0	1	1	1
***Oedogonium* sp. CQ1923_8**	2	1	0	0	1	2	1	1	3
***Oedogonium dentireticulatum* CQ1925_1**	2	1	1	1	0	0	1	1	1
***Oedogonium* sp. CQ1931**	3	1	0	0	0	0	1	0	1
***Oedogonium* sp. CQ1932_4**	2	1	1	1	0	0	1	1	0
***Oedogonium* sp. CQ1933_1**	2	1	0	1	1	2	1	1	1
***Oedogonium* sp. CQ1940_1**	0	1	0	0	0	0	1	2	1
***Oedogonium* sp. CQ1940_2**	2	1	1	1	1	2	1	1	2
***Oedogonium* sp. FYT1801**	0	1	0	0	0	0	1	2	1
***Oedogonium* sp. GDQY3**	2	1	1	1	0	0	1	1	2
***Oedogonium* sp. HH1810_2**	0	0	0	0	0	0	1	2	1
***Oedogonium* sp. SD1809**	0	0	0	0	0	0	1	2	1
***Oedogonium* sp. HH1815_2**	0	0	0	0	0	0	1	2	1
***Oedogonium varians* TS191012**	1	1	1	1	1	2	1	1	2
***Oedogonium* sp. TS1907_1**	1	1	2	0	1	2	1	1	1
***Oedogonium* sp. TS1907_2**	3	1	1	2	0	0	1	1	0
***Oedogonium* sp. TS1907_4**	0	2	1	2	0	0	1	1	0
***Oedogonium obpyriforme* TS1910_7**	2	2	1	2	1	2	1	1	3
***Oedogonium* sp. WD1801**	0	0	0	0	0	0	1	2	0
***Oedogonium capilliforme* WH1904**	2	1	1	1	1	1	1	1	2
***Oedogonium* sp. WH1911_4**	0	1	0	0	0	0	1	1	0
***Oedogonium* sp. WH1913_5**	0	1	0	0	0	0	1	1	0
***Oedogonium* sp. WH1913_6**	0	2	1	2	0	0	1	1	2
***Oedogonium* sp. WH1914_1**	1	1	2	1	1	2	1	1	0
***Oedogonium* sp. WH1920_1**	2	1	2	1	0	0	1	0	0
** *Oedogonium* ** ***crispum* var. *hawaiense* WH1924_1**	1	1	1	0	0	0	1	1	0
***Oedogonium* sp. WH1924_2**	3	1	2	1	0	0	1	1	0
***Oedogonium* sp. WH1924_3**	0	1	1	1	0	0	1	1	0
***Oedogonium mirificum* WH1931_10**	3	1	2	2	0	0	1	1	1
***Oedogonium mirificum* WH1931_11**	3	1	2	2	0	0	1	1	1
***Oedogonium mirificum* WH1931_6**	3	1	2	2	0	0	1	1	1
***Oedogonium cyathigerum* WH1931_8**	3	2	1	2	0	0	1	1	1
***Oedogonium* sp. WLQ1851006_1_2**	0	0	0	0	0	0	1	2	1
***Oedogonium* sp. WLQ1851010_1**	0	0	0	0	0	0	1	2	1
***Oedogonium* sp. WLQ1852911_4_4**	2	1	1	1	0	0	1	0	0
***Oedogonium capillare* XQ1802**	2	1	1	1	2	2	1	1	0
***Oedogonium* sp. XQ1804**	0	0	0	0	0	0	1	2	1
***Oedogonium* sp. XQ1819**	0	0	0	0	0	0	1	2	1
***Oedogonium* sp. XQ1825**	0	1	1	1	0	0	1	0	0
***Oedogonium* sp. XT1902_1**	1	1	0	2	1	2	1	1	0
***Oedogonium* sp. XT1902_3**	1	1	1	2	1	2	1	1	0
***Oedogonium* sp. XT1902_5**	0	1	0	0	0	0	1	2	0
***Oedogonium* sp. ZWY1801_6**	0	0	0	0	0	0	1	2	1
***Oedogonium* sp. ZWY1805**	0	1	2	1	0	0	1	0	0
*Oedogonium* sp. ZWY1903_1	0	1	0	0	0	0	1	2	0
*Oedogonium* sp. FACHB990	2	0	0	0	0	0	1	0	0
*Oedogonium* sp. FACHB991	1	0	0	0	0	0	1	0	0
*Oedogonium* sp. FACHB992	1	0	0	0	0	0	1	0	0
*Oedogonium* sp. FACHB994	1	0	0	0	0	0	1	0	0
*Oedogonium* sp. FACHB998	3	0	0	0	0	0	1	0	0
*Oedogonium subplagiostomum* Ley FACHB989	2	1	1	1	1	2	1	1	0
*Oedogonium brevicingulatum* Jao FACHB999	1	1	1	1	1	2	1	1	2
*Oedogonium nodulosum* Wittr. FACHB996	1	1	2	1	1	2	2	1	1
*Oedogonium borisianum* (Le Clerc) Wittr. UTEX LB2239	3	2	1	1	0	1	2	1	3
*Oedogonium calliandrum* Hoffm. UTEX LB1554	2	1	1	1	1	2	1	1	1
*Oedogonium cardiacum* Wittr. UTEX LB40	2	1	1	1	1	2	1	1	1
*Oedogonium angustistomum* Hoffmann UTEX LB1557 UTEX LB1557	2	1	1	1	1	2	1	1	1
*Oedogonium vaucherii* (Le Clerc) A. Braun, SCCAP K0094	1	1	1	1	1	2	1	1	2
*Oedogonium fragile* Wittr. SCCAP K0093	1	1	1	1	1	2	1	1	0
*Oedogonium crispum* (Hass.) Wittr. ACOI 1287	1	1	2	1	1	2	1	1	1
*Oedogonium oblongum* Wittr.,ACOI 1118	1	2	2	1	1	2	1	1	1/3
*Oedogonium cylindrosporum* Jao	3	2	2	2	1	2	1	1	1/3
*Oedogonium eminens* (Hirn) Tiff.	3	1	2	2	0	1	2	1	1
DQ178024 *Oedogonium tenerum*	2	1	1	1	1	2	1	1	1
DQ178025 *Oedogonium undulatum* (Brébisson) A. Braun	3	1	2	1	0	1	2	1	1
*Oedogonium pakistanense* Islam & Sarma FACHB995	1	1	2	1	1	2	1	0	1
KY575148 *Oedogonium subplagiostomum*	2	1	1	1	1	2	1	1	0

1. Sex differentiation: monoecious species (1); dioecious, macrandrous species (2); dioecious, nannandrous, gynandrosporous species (3); dioecious, nannandrous, idioandrosporous (4). 2. The shape of oogonium: globose or subglobose (1); ellipsoid (2). 3. Type of oogonium aperture: pore (1); circumcision (2). 4. Oospore wall ornamentation: smooth (1); with ornamentations (2). 5. The division way of antheridium: horizontal division (1); vertical division (2). 6. The number of sperm of each antheridium: 1 (1); 2 (2). 7. Vegetative cell: cylindrical or subcylindrical (1); noncylindrical (2). 8. Basal cell: elongated (1); subhemispherical or nearly spherical (2). The strains newly collected in this study are marked in bold.

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
