# Peer review of "Morphology and Molecular Phylogeny of Genus Oedogonium (Oedogoniales, Chlorophyta) from China"

_plants, 2022, doi:10.3390/plants11182422_

Round 1

Reviewer 1 Report

The manuscript is not good but okay. The English should really be checked by a native. Here and there the manuscript seems illogical. The authors use once the ITS region (ITS1+5.8S+ITS2) and another time a concatenated dataset of 18S + the ITS region + rbcL. Either they should present only the concatenated dataset (if they trust that more = better?) or they should present each marker separately - i.e. 18S, the ITS region, rbcL, and then, as an addition, the concatenated dataset. This way they could really discuss where individual markers contradict or agree - also regarding morphological features. The same is actually true for the ITS region. It is known that ITS2 alone works sometimes better than the whole ITS region. Also here it would make sense to consider the markers ITS1, ITS2 and 5.8S separately and then the concatenated data for comparison! If this is all too much work, then, as I said, perhaps only the concatenated dataset should be considered. However, I myself am a friend of single-marker analyses. Due to manual corrections in the alignments, it is mandatory to store the alignments as a supplement. It is known that the ITS region can hardly be aligned without knowledge of the secondary structures. For ITS2 (and also for the 18S) oedogonialean secondary structures are well known. They could serve as a guide, or sequence-structure data could be used together to achieve better results. At least, the authors should provide a statement, if the ITS region could easily be aligned? In summary, the manuscript, as mentioned above, is not really good but okay; I agree with the currently presented results.

Author Response

Reviewer 1

The manuscript is not good but okay. The English should really be checked by a native. Here and there the manuscript seems illogical.

Re: Thank you very much for your proposal. We have checked the expression in the manuscript and the revisions have been made.

The authors use once the ITS region (ITS1+5.8S+ITS2) and another time a concatenated dataset of 18S + the ITS region + rbcL. Either they should present only the concatenated dataset (if they trust that more = better?) or they should present each marker separately - i.e. 18S, the ITS region, rbcL, and then, as an addition, the concatenated dataset. This way they could really discuss where individual markers contradict or agree - also regarding morphological features.

Re: Thank you very much for your suggestion. In this study, we used the concatenated dataset to reconstruct the phylogenetic relationship of the Oedogonium and Oedocladium considering the poor support value of the phylogenetic result based on the individual markers. And we also found the general topology based on ITS sequence was similar with the result by concatenated dataset, since the topology by ITS sequence included more species with complete morphological characteristics, then we used the phylogenetic tree by ITS combined with morphological studies to search the morphological characteristics which could reflect the phylogeny of Oedogonium.

The same is actually true for the ITS region. It is known that ITS2 alone works sometimes better than the whole ITS region. Also here it would make sense to consider the markers ITS1, ITS2 and 5.8S separately and then the concatenated data for comparison! If this is all too much work, then, as I said, perhaps only the concatenated dataset should be considered.

Re: Thank you for the useful suggestions. As advided, we tried to use the ITS2 sequence with 213bp to reconstruct the phylogenetic tree, and the topology was consistent with the topology of the concatenated dataset by ITS1, ITS2 and 5.8S. Since the length of the concatenated dataset was 375bp, and the included ITS1 and 5.8S were very conserved, we may think the concatenated dataset can be used to be the marker with no doubt. And the phylogenetic tree based on the ITS2 as shown followed:

However, I myself am a friend of single-marker analyses. Due to manual corrections in the alignments, it is mandatory to store the alignments as a supplement. It is known that the ITS region can hardly be aligned without knowledge of the secondary structures. For ITS2 (and also for the 18S) oedogonialean secondary structures are well known. They could serve as a guide, or sequence-structure data could be used together to achieve better results. At least, the authors should provide a statement, if the ITS region could easily be aligned? In summary, the manuscript, as mentioned above, is not really good but okay; I agree with the currently presented results.

Re: Thank you for the useful suggestions. In this study, the ITS (ITS1+5.8S+ITS2) aligned well after manual corrections. ITS2 secondary structures may be useful for the species with great variation in the ITS sequence, but in the Oedogoniales, the ITS2 sequence was relatively conserved than other groups in Chlorophyceae, such as in Scenedesmaceae. We also have tried to analyze the consensus ITS2 secondary structure model of Oedogonium, which also indicated the relative conservation of the ITS2. Since the phylogenetic results based on the 18S+ITS+rbcl, ITS (ITS1+5.8S+ITS2) and ITS2 (in the figure above) showed the same result that genus Oedogonium can be divided into two clades, with the Clade II species showed spherical or sub-hemispherical basal cells, we may think these have achieved the purpose of this study. And we will upload the alignment result as the supplementary file.

Reviewer 2 Report

This is an interesting and well-written paper.  My only major suggestion is that when the authors suggest that Oedegonium is polyphyletic they mean paraphyletic. Polyphyletic means that the species classified as Oedogonium were not all derived from a common ancestor classified as Oedogonium. The cladograms do not show this. What the cladograms show is that Oedogonium is monophyletic but also paraphyletic, i.e. some members of the clade belong to the genera Oeodocladium and Bulbochaete.

Other comments.

Lines 49-50. Change to ‘monoecious; dioecious, macrandrous; dioecious, nannandrous’

Line 115. ‘composed of’, not ‘constituted by’.

Lines 133-134. Change to ‘monoecious; dioecious, macrandrous; dioecious, nannandrous’.

Lines 140, 141. ‘possessed, not ‘presented with’.

Line 175. ‘reflect’, not ‘reflecting’.

Author Response

Reviewer 2

This is an interesting and well-written paper. My only major suggestion is that when the authors suggest that Oedegonium is polyphyletic they mean paraphyletic. Polyphyletic means that the species classified as Oedogonium were not all derived from a common ancestor classified as Oedogonium. The cladograms do not show this. What the cladograms show is that Oedogonium is monophyletic but also paraphyletic, i.e. some members of the clade belong to the genera Oeodocladium and Bulbochaete.

Re: L32, 33, 47, 102, 170-172. Thank you very much for the suggestion. We really agree with you, and this kind of expression has been revised in the newly manuscript.

Other comments.

Lines 49-50. Change to ‘monoecious; dioecious, macrandrous; dioecious, nannandrous’

Line 115. ‘composed of’, not ‘constituted by’.

Lines 133-134. Change to ‘monoecious; dioecious, macrandrous; dioecious, nannandrous’.

Lines 140, 141. ‘possessed, not ‘presented with’.

Line 175. ‘reflect’, not ‘reflecting’.

Re: Thank you very much for your advice. These spelling or expression mistakes have been corrected, and we also check such problems in the manuscript.

Round 2

Reviewer 1 Report

I can follow and understand the authors' reply. However, I would like to have these "gene-marker" concatenations/choices/decissions/discussions/comparisons also included in the manuscript, not only in the reply letter. Further, language is still an issue, e.g. the following is just one sentence from the manuscript that no one can understand:

"Phylogenetic analysis combined with morphological studies revealed the types of sexual differentiation, the characters of oogonium (including the shape, the types of the aperture and the ornamentation of oospore wall), the division types of antheridial and the number of sperm of each antheridial which were considered as the most important morphological character in the traditional taxonomy of Oedogonium did not form monophyletic lineages respectively, which implied the traditional morphological character using for taxonomy of Oedogoniales may not reflect the real phylogeny of this group."

Author Response

Reviewer 1

I can follow and understand the authors' reply. However, I would like to have these "gene-marker" concatenations/choices/decissions/discussions/comparisons also included in the manuscript, not only in the reply letter.

Re: L135-137, 144-147,154-157. Thank you very much for your proposal. We have revised this part in the manuscript.

Further, language is still an issue, e.g. the following is just one sentence from the manuscript that no one can understand: "Phylogenetic analysis combined with morphological studies revealed the types of sexual differentiation, the characters of oogonium (including the shape, the types of the aperture and the ornamentation of oospore wall), the division types of antheridial and the number of sperm of each antheridial which were considered as the most important morphological character in the traditional taxonomy of Oedogonium did not form monophyletic lineages respectively, which implied the traditional morphological character using for taxonomy of Oedogoniales may not reflect the real phylogeny of this group."

Re: L26, 27, 92, 160-162, 192-202, 227. Thank you very much for the advice. We have revised the language again.